# Cross-sectional study on the prevalence of influenza and pneumococcal vaccination and its association with health conditions and risk factors among hospitalized multimorbid older patients

**Dimitrios David Papazoglou**[1,2]*, **Oliver Baretella**[1,2], **Martin Feller**[1], **Cinzia Del Giovane**[1,3],
**Elisavet Moutzouri**[1,2], **Drahomir Aujesky**[2], **Matthias Schwenkglenks**[4],
**Denis O'Mahony**[5,6], **Wilma Knol**[7], **Olivia Dalleur**[8], **Nicolas Rodondi**[1,2],
**Christine Baumgartner**[2]

1 Institute of Primary Health Care (BIHAM), University of Bern, Bern, Switzerland, 2 Department of General Internal Medicine, Inselspital, Bern University Hospital, Bern, Switzerland, 3 Population Health Laboratory, University of Fribourg, Fribourg, Switzerland, 4 Institute of Pharmaceutical Medicine (ECPM), University of Basel, Basel, Switzerland, 5 Department of Medicine Cork, University College Cork National University of Ireland, Munster, IE, Ireland, 6 Department of Geriatric Medicine Cork, Cork University Hospital Group, Munster, IE, Ireland, 7 Department of Geriatrics and Expertise Centre Pharmacotherapy in Old Persons (EPHOR), University Medical Center Utrecht, Utrecht University, Utrecht, The Netherlands, 8 Louvain Drug Research Institute, and Pharmacy Department, Cliniques Universitaires Saint-Luc, Université Catholique de Louvain, Brussels, Belgium

* papazoglou2019@gmail.com

**Data Availability Statement:** Data for this study will be made available for scientific purposes upon

## Abstract

### Background

Older adults with chronic conditions are at high risk of complications from influenza and pneumococcal infections. Evidence about factors associated with influenza and pneumococcal vaccination among older multimorbid persons in Europe is limited. The aim of this study was to investigate the prevalence and determinants of these vaccinations in this population.

### Methods

Multimorbid patients aged ≥70 years with polypharmacy were enrolled in 4 European centers in Switzerland, Belgium, the Netherlands, and Ireland. Data on vaccinations, demographics, health care contacts, and comorbidities were obtained from self-report, general practitioners and medical records. The association of comorbidities or medical contacts with vaccination status was assessed using multivariable adjusted log-binomial regression models.

### Results

Among 1956 participants with available influenza vaccination data (median age 79 years, 45% women), 1314 (67%) received an influenza vaccination within the last year. Of 1400

request for researchers whose proposed use of the data has been approved by the OPERAM publication committee. After approval and signing of a data transfer agreement ensuring adherence to privacy and data handling, data and documentation will be made available through a secure file exchange platform. Partially deidentified participant data, a data dictionary and annotated case report forms will be made available. For data access, external researchers can contact operam@biham. unibe.ch.

**Funding:** This work is part of the project "OPERAM: OPtimising thERapy to prevent Avoidable hospital admissions in the Multimorbid elderly" supported by the European Union's Horizon 2020 research and innovation program under the grant agreement No 634238, and by the Swiss State Secretariat for Education, Research and Innovation (SERI) under contract number 15.0137. The opinions expressed and arguments employed herein are those of the authors and do not necessarily reflect the official views of the European Commission and the Swiss government. This project was also partially funded by the Swiss National Scientific Foundation (SNSF 320030_188549). The funder of the study had no role in study design, data collection, data analysis, data interpretation or writing of the report.

**Competing interests:** The authors have declared that no competing interests exist.

patients with available pneumococcal vaccination data (median age 79 years, 46% women), prevalence of pneumococcal vaccination was 21% (n = 291). The prevalence of vaccination remained low in high-risk populations with chronic respiratory disease (34%) or diabetes (24%), but increased with an increasing number of outpatient medical contacts. Chronic respiratory disease was independently associated with the receipt of both influenza and pneumococcal vaccinations (prevalence ratio [PR] 1.09, 95% confidence interval [CI] 1.03–1.16; and PR 2.03, 95%CI 1.22–3.40, respectively), as was diabetes (PR 1.06, 95%CI 1.03–1.08; PR 1.24, 95%CI 1.16–1.34, respectively). An independent association was found between number of general practitioner visits and higher prevalence of pneumococcal vaccination (p for linear trend <0.001).

## Conclusion

Uptake of influenza and particularly of pneumococcal vaccination in this population of European multimorbid older inpatients remains insufficient and is determined by comorbidities and number and type of health care contacts, especially outpatient medical visits. Hospitalization may be an opportunity to promote vaccination, particularly targeting patients with few outpatient physician contacts.

## Introduction

Seasonal influenza is a contagious viral respiratory disease, which can cause mild to severe illness. Influenza affects approximately 4 to 50 million individuals and causes 15,000 to 70,000 deaths in the European Union (EU) each year [1]. Older persons and those with chronic medical conditions are most vulnerable of developing associated complications, which can lead to hospitalization and death [2, 3]. Health authorities from almost all EU countries recommend the influenza vaccination for individuals over 65 years of age independently of their comorbidities [4].

Manifestations of pneumococcal disease caused by infection with Streptococcus pneumoniae include community-acquired pneumonia, meningitis, as well as severe invasive pneumococcal disease. Pneumococcal disease is associated with high morbidity and mortality and is responsible for a larger number of deaths worldwide than influenza or HIV [5, 6]. The group most frequently affected by pneumococcal disease are children <2 years of age and adults older than 65 years, who are also at particularly high risk of associated mortality [7]. The pneumococcal vaccination is cost-effective in older individuals and can reduce invasive pneumococcal disease, pneumonia, and hospitalizations from pneumonia [8–13]. At present, most but not all European countries recommend pneumococcal vaccination for all individuals over 65 years of age independently of their comorbidities [14]. In other countries like Switzerland, pneumococcal immunization is recommended for patients with underlying diseases predisposing to invasive pneumococcal disease or at high risk for complications, for example those with chronic respiratory, cardiovascular, or kidney diseases, organ transplants, or haematologic malignancies [15].

Despite the known effectiveness of influenza and pneumococcal vaccination as a simple measure to reduce these diseases [16, 17], and the 75% European Centre for Disease Prevention and Control (ECDC) influenza vaccination coverage target for the EU in older people by 2015, actual coverage is much lower at around 45% [18]. Most studies examining factors

associated with influenza and pneumococcal vaccination were conducted in Asia and America [19–21], while fewer data exist on older individuals in European countries. Data on pneumococcal vaccination status in adults in Europe is particularly scarce [22]. Furthermore, we are aware of only a few smaller studies examining inappropriate lack of vaccination specifically among hospitalized multimorbid older patients who are at particularly high risk of subsequent influenza- and pneumococcal-associated morbidity and mortality [23–25]. Identifying determinants of inappropriate lack of vaccination in this population is crucial for targeting public health interventions and reducing the burden of these potentially preventable diseases.

The aim of our study was to investigate the prevalence of influenza and pneumococcal vaccination in a population of multimorbid older inpatients and its determinants, taking advantage of baseline data from the OPERAM (OPtimising thERapy to prevent Avoidable hospital admissions in Multimorbid older people) trial [26].

## Materials and methods

### Study design and participants

We conducted a cross-sectional study using baseline data from the OPERAM trial (Clinical-Trials.gov Identifier: NCT02986425). Details of the study design have been previously published [26]. In brief, OPERAM is a European multicenter cluster-randomized controlled trial investigating whether a medication optimization intervention during hospitalization compared to usual care can reduce the risk of drug-related hospital admissions among multimorbid older patients with polypharmacy. Overall, 2008 participants were recruited between December 2016 and October 2018 in four European university hospital centers in Switzerland, Belgium, the Netherlands, and Ireland. To be eligible for the OPERAM trial, patients had to be ≥70 years of age, multimorbid (defined as having ≥3 chronic health conditions) and had to take five or more chronic medications. Patients who were directly admitted to palliative care or those with a structured drug review within two months prior to screening were excluded from participation. We considered all OPERAM participants with available baseline data on vaccination status for the current study.

The local ethics committee at each of the participating sites approved the protocol of the OPERAM study (Cantonal Ethics Committee Bern, Switzerland; Medical Research Ethics Committee Utrecht, Netherlands; Comité d'Ethique Hospitalo-Facultaire Saint-Luc-UCL, Belgium; Cork University Teaching Hospitals Clinical Ethics Committee, Ireland). All participants or their legal representatives provided written informed consent.

### Variables

**Influenza and pneumococcal vaccination.** All data from the OPERAM trial were entered into a central database. To obtain information on vaccinations, the patient or next of kin were asked if and when the last influenza and pneumococcal vaccination was given. As the influenza vaccination needs to be administered annually, influenza vaccination status was coded as "yes" only if the influenza vaccination had been administered within 12 months before study inclusion. For the analysis of pneumococcal vaccination, all participants from the Netherlands were excluded as this information was not routinely collected.

**Covariates.** Data on patient characteristics (age, sex, race, highest level of education, smoking status, and alcohol consumption) and use of medical resources (hospitalizations within the last year, visits to a general practitioner [GP], specialist, emergency department [ED], or hospital outpatient clinic within the last 6 months, nursing home stays or home nursing visits, or receipt of informal care [i.e., unpaid care by family members, relatives or friends] during the last 6 months) were collected at inclusion from the patient or next of kin.

We categorized the patients in clinical risk groups prone to a complicated or fatal course of influenza and pneumococcal infection. Definitions were taken from the infectious diseases "green book", the immunization recommendations in the UK published by Public Health England, describing detailed clinical risk groups [27, 28]. These include patients with chronic respiratory disease, chronic heart disease, chronic kidney disease, chronic liver disease, diabetes, immunosuppression, and individuals with asplenia or dysfunction of the spleen, cochlear implants, or cerebrospinal fluid leaks. These clinical risk groups were coded using the International Statistical Classification of Diseases and Related Health Problems 10th revision (ICD-10) for health conditions and the Anatomical Therapeutic Chemical (ATC) Classification codes for medications as previously suggested [29–32]. ICD-10 coded health conditions were obtained from discharge reports and only diagnoses with a diagnosis date less recent than the admission date were included. Information on medications using ATC codes was obtained from medical records, patient interview or pharmacy or GP lists. Medication data were used for the definition of diabetes and immunosuppression. Immunosuppression was defined as intake of at least 20mg of prednisone equivalent daily for more than a month and diabetes as the intake of any glucose-lowering agent as well as a corresponding ICD-10 code for diabetes mellitus. Due to the extremely low numbers of patients with asplenia or dysfunction of the spleen, cochlear implants, or cerebrospinal fluid leaks in our population, we did not investigate these clinical risk groups in our study.

The Charlson Comorbidity Index (CCI) to predict 10-year survival was calculated. The index ranges from 0 to 33, with lower scores indicating higher 10-year survival [33]. Quality of life was assessed using the European Quality of Life-5 Dimensions (EQ-5D) instrument. EQ-5D as a measure of generic, self-reported health status was administered following the rules of the EuroQol consortium. Country specific value sets were used to translate questionnaire responses to a health states measure on a 1 to 0 scale, with a value of 1 corresponding to perfect health and a value of 0 to death [34–36]. The German value set was used for the Swiss study site in Bern and the French for the study site in Louvain, Belgium, due to missing value sets for these countries.

## Statistical methods

We compared patient characteristics by receipt of influenza and pneumococcal vaccination using chi squared tests for categorical variables, and Student's t-tests or Wilcoxon rank sum tests for continuous variables, as appropriate. Prevalence of influenza and pneumococcal vaccination overall and according to patient characteristics were calculated along with corresponding 95% confidence intervals (CI). To assess the association of chronic health conditions or use of medical resources with vaccination status, we used log-binomial regression models with vaccination status as the dependent variable and health conditions and variables for medical resource utilization as the main independent variables in separate models to compute unadjusted prevalence rate ratios and 95% CIs. We used log-binomial regression models because this is the recommended method to estimate prevalence ratios [37, 38]. We then adjusted these models for age, sex, race, education, alcohol use, and smoking status to investigate whether or not health conditions and medical resource utilization were associated with vaccination status independent of these variables. Furthermore, we clustered the analyses by study sites to take into account the participants' correlation within each site, thus allowing for intragroup correlation of standard errors.

We carried out sensitivity analyses using Poisson regression models to assess the robustness of our results [37]. An additional sensitivity analysis for pneumococcal vaccination was done excluding patients from Switzerland, where national pneumococcal vaccination guidelines

differ from those in the other two countries for which data on pneumococcal vaccination was available for our study participants. For these countries the pneumococcal vaccination is recommended in all patients aged 65 years or older; in Switzerland, the recommendation only applies to clinical risk groups with advanced chronic health conditions independent of their age (e.g. from COPD GOLD III, NYHA III, KDIGO G4) [15, 28, 39, 40].

We performed all statistical analyses using STATA version 13.1 (StataCorp, College Station, TX). Two-sided p-values of 0.05 were considered statistically significant. The STROBE statement was used for reporting this cross-sectional study [41].

## Results

Of 2008 patients who were included in the OPERAM trial, 1956 multimorbid inpatients with available vaccination data were included in the analysis on influenza vaccination, and 1400 participants from three study sites were included in the analysis on pneumococcal vaccination. Of the 52 patients without available influenza data, 7 withdrew, and in 45 patients vaccination status was unknown. For the analysis of pneumococcal vaccination, all 452 participants from the Netherlands had to be excluded because this information had not been routinely collected at this study site. Of the remaining 1556 patients, 9 withdrew and information was lacking for 147 patients (Fig 1).

Table 1 shows baseline characteristics of the study population by influenza and pneumococcal vaccination status. Overall, the median age was 79 years [IQR 74–84] and 45% were women.

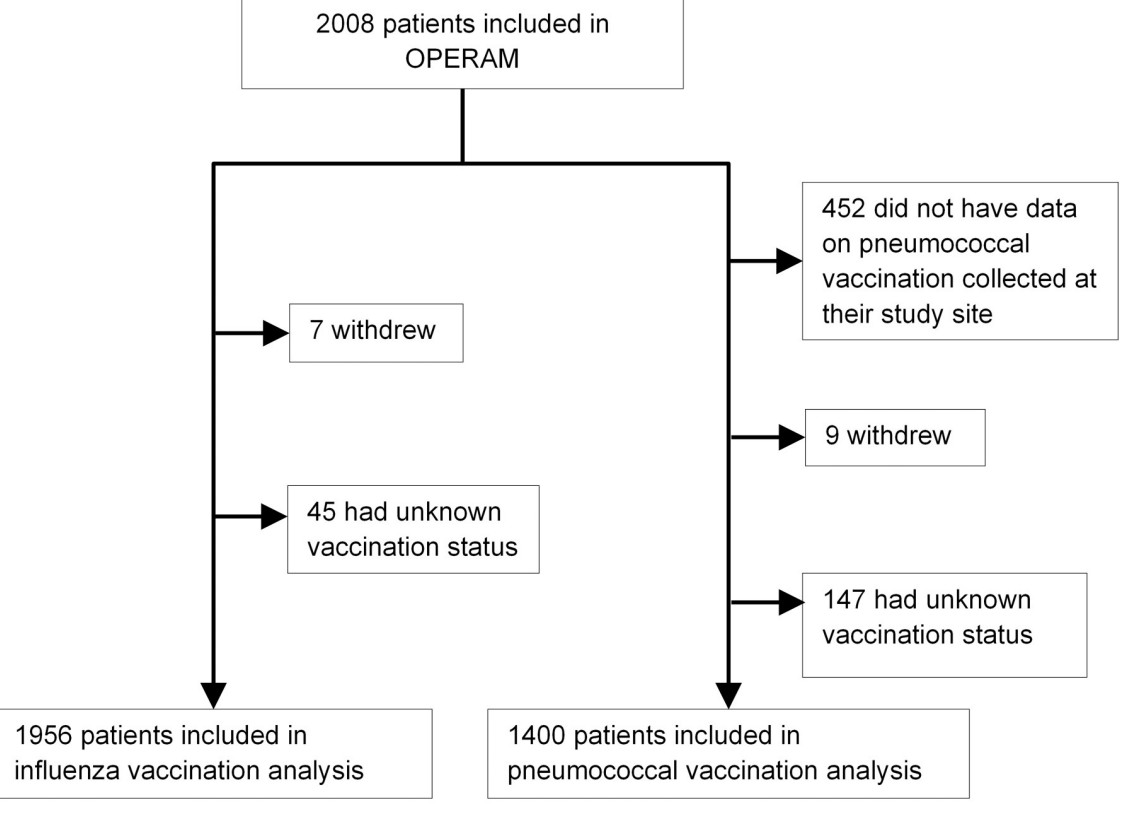

**Fig 1. Overview of the study population.**

**Table 1. Patient characteristics according to vaccination status.**

| Characteristics | Influenza Vaccination | | | Pneumococcal Vaccination | | |
|---|---|---|---|---|---|---|
| | Yes (n = 1,314) | No (n = 642) | p-value‡ | Yes (n = 291) | No (n = 1109) | p-value‡ |
| Female sex | 589 (44.8) | 289 (45.0) | 0.94 | 154 (52.9) | 491 (44.3) | 0.008 |
| Age, years | 79 (75–85) | 77 (73–82) | <0.001 | 79 (74–84) | 79 (74–84) | 0.52 |
| Education | | | 0.011 | | | <0.001 |
| Less than high school | 413 (31.7) | 166 (26.2) | | 123 (42.6) | 292 (26.7) | |
| High school | 571 (44.0) | 321 (50.7) | | 81 (28.0) | 514 (46.9) | |
| University | 319 (24.0) | 146 (23.0) | | 85 (29.4) | 289 (26.4) | |
| Current smoker | 87 (6.6) | 66 (10.3) | 0.005 | 19 (6.5) | 88 (8.0) | 0.42 |
| Alcohol SD per week | 0 (0–4) | 0 (0–3) | 0.59 | 0 (0–2) | 0 (0–3.5) | 0.30 |
| **Health Care Contacts*** | | | | | | |
| GP visits, n | | | <0.001 | | | <0.001 |
| 0 | 80 (6.2) | 57 (9.0) | | 6 (2.1) | 77 (7.0) | |
| 1–2 | 357 (27.5) | 198 (31.3) | | 67 (23.3) | 325 (29.7) | |
| 3–4 | 316 (24.3) | 167 (26.4) | | 71 (24.7) | 269 (24.6) | |
| ≥ 5 | 547 (42.1) | 211 (33.3) | | 144 (50.0) | 423 (38.7) | |
| Other outpatient physician or ED visits, n | | | 0.024 | | | 0.069 |
| 0 | 413 (31.8) | 246 (38.6) | | 101 (35.2) | 423 (38.5) | |
| 1–2 | 507 (39.0) | 217 (34.0) | | 106 (36.9) | 435 (39.6) | |
| ≥ 3 | 380 (29.2) | 175 (27.4) | | 80 (27.9) | 241 (21.9) | |
| Hospitalizations, n | | | 0.71 | | | 0.95 |
| 0 | 641 (48.9) | 316 (49.5) | | 146 (50.2) | 543 (49.2) | |
| 1 | 380 (29.0) | 167 (26.2) | | 79 (27.2) | 325 (29.4) | |
| ≥ 2 | 291 (22.2) | 155 (24.3) | | 66 (22.7) | 236 (21.4) | |
| Nursing home resident | 129 (9.9) | 54 (8.4) | 0.31 | 29 (10.0) | 108 (9.8) | 0.90 |
| Any home nursing visits | 346 (26.5) | 157 (24.5) | 0.36 | 80 (27.6) | 266 (24.1) | 0.22 |
| Receipt of informal care† | 312 (23.9) | 112 (17.5) | 0.001 | 72 (24.8) | 194 (17.6) | 0.005 |
| **Health Indexes** | | | | | | |
| EQ-5D§ | 0.89 (0.65–1) | 0.89 (0.70–1) | 0.44 | 0.86 (0.57–1) | 0.89 (0.68–1) | 0.05 |
| CCI** | 6 (5–7) | 5 (4–7) | <0.001 | 6 (5–7) | 5 (4–7) | 0.10 |
| **Clinical risk groups** | | | | | | |
| Chronic heart disease | 682 (52.0) | 319 (49.7) | 0.34 | 136 (49.9) | 593 (53.5) | 0.044 |
| Chronic respiratory disease | 377 (28.7) | 120 (18.7) | <0.001 | 104 (35.9) | 198 (17.9) | <0.001 |
| Chronic liver disease | 67 (5.1) | 44 (6.9) | 0.12 | 19 (6.6) | 77 (7.0) | 0.81 |
| Chronic kidney disease | 116 (8.8) | 34 (5.3) | 0.006 | 26 (9.0) | 87 (7.9) | 0.54 |
| Diabetes mellitus | 444 (33.8) | 202 (31.5) | 0.29 | 107 (36.9) | 345 (31.1) | 0.06 |
| Rheumatic disease | 109 (8.3) | 51 (7.9) | 0.78 | 28 (9.7) | 86 (7.8) | 0.29 |
| Any malignancy†† | 337 (25.7) | 159 (24.8) | 0.66 | 69 (23.8) | 234 (21.1) | 0.33 |
| Immunosuppression | 106 (8.1) | 56 (8.7) | 0.63 | 31 (10.7) | 82 (7.4) | 0.07 |

Numbers are presented as n (%), or median (interquartile range). The number of missing data in participants included in the analysis on influenza / pneumococcal vaccination, respectively, was: education n = 29 / n = 27, ethnicity n = 6 / n = 5, smoking n = 10 / n = 8, alcohol n = 17 / n = 15, GP visits n = 18 / n = 15, other outpatient physician or ED visits n = 17 / n = 15, hospitalization n = 13 / n = 10, nursing home resident n = 10 / n = 8, any home nursing visits n = 16 / n = 14, receipt of informal care n = 15 / n = 13, and n = 4 /n = 3 for all of the chronic health conditions defining the clinical risk groups. Abbreviations: CCI, Charlson comorbidity index; ED, emergency department; GP, general practitioner; SD, standard drinks.

‡ In case of GP visits, other outpatient physician or ED visits, and hospitalizations, the p-value refers to a p for trend.

* Health care contacts refer to hospitalizations within 12 months, or GP visits, ED or outpatient clinic/specialist visits, receipt of informal care, any nursing home visits, or permanent nursing home residency within 6 months prior to the baseline visit.

† defined as care received by relatives or other close persons.

§ Questionnaire-based health status on a 1 to 0 scale. A value of 1 corresponds to perfect health and a value of 0 to death".

** The CCI predicts 10-year survival in patients with multiple comorbidities and ranges from 0 to 33 points. Lower scores indicate a higher risk of 10-year-survival. 7 points correspond to an estimated 0% 10-year survival.

†† Except malignant neoplasm of skin.

## Influenza vaccination

Overall, 1314 of 1956 (67.2%) hospitalized multimorbid older patients were vaccinated against influenza within one year before study inclusion (Table 1). Median age was higher in the group with a positive compared to those with a negative influenza vaccination status (median age 79 years [IQR 75–85] vs. 77 years [IQR 73–82], while the percentage of women did not differ between the two groups (Table 1). Those who received an influenza vaccination were less likely to smoke (6.6% vs. 10.3%).

The probability of having received an influenza vaccination increased with age and with an increasing number of GP visits (p for trend both <0.001, Table 2). The prevalence of influenza vaccination was higher in individuals with low (less than high school, 71.3%) and high education levels (University, 68.6%) compared to those with an intermediate education level (high school, 64.0%). Above average influenza vaccination rates were seen in those with more comorbidities (CCI ≥ 7; 72.8%), patients receiving informal care within the last 6 months (73.6%) and in those with chronic respiratory (75.9%) and chronic kidney disease (77.3%).

In multivariable analysis, presence of chronic respiratory disease, chronic kidney disease and diabetes mellitus were independently associated with a higher prevalence of influenza vaccination, with adjusted prevalence ratios (PR) of 1.09 (95% CI 1.03–1.16), 1.12 (95% CI 1.08–1.17), and 1.06 (95% CI 1.03–1.08), respectively, while immunosuppression showed no association (Table 3). Specialist or ED visits in the last six months (PR 1.12, 95% CI 1.01–1.24 for ≥3 compared to no visit, p for linear trend 0.027) as well as a CCI ≥ 7 (PR 1.10, 95% CI 1.02–1.19) were associated with the receipt of influenza vaccination (Table 3).

The results were similar in unadjusted analyses only considering clinical risk groups, health care contacts and health scores (S1 Table) and sensitivity analyses using Poisson regression models (S2 Table).

## Pneumococcal vaccination

In the population of hospitalized multimorbid older patients with polypharmacy from 3 study sites (n = 1400), 20.8% (n = 291) were vaccinated against pneumococcal disease (Table 1). Patients with a pneumococcal vaccination were more likely to be female (52.9% vs. 44.3%) and to have a lower level of education compared to those without the vaccination (less than high school degree 42.6% vs. 26.7%), while age and smoking status were similar in both groups (Table 1).

The prevalence of pneumococcal vaccination was higher with an increasing number of GP visits and in those with ≥3 compared to those with <3 other outpatient physician or ED visits in the last 6 months (Table 2). Above average pneumococcal vaccination rates were seen in those receiving informal care (27.1%), those with chronic respiratory disease (34.3%), immunosuppression (27.4%), rheumatic disease (24.6%), and diabetes mellitus (23.7%).

A diagnosis of chronic respiratory disease, diabetes, or immunosuppression was associated with a higher prevalence of pneumococcal vaccination in multivariable adjusted analyses (PR 2.03, 95% CI 1.22–3.40; PR 1.24, 95% CI 1.16–1.34; and PR 1.29, 95% CI 1.03–1.61, respectively). Most other chronic health conditions investigated in our analysis including chronic heart, chronic liver or rheumatic disease and cancer, showed no association with pneumococcal vaccination, similar to the findings for influenza vaccination. GP visits within the last 6 months were very strongly associated with the receipt of pneumococcal vaccination (p for linear trend <0.001), showing a more than threefold higher prevalence of vaccination (PR 3.41, 95% CI 2.74–3.72) in those with 5 or more visits compared to those without GP visits within the last 6 months. The results were similar in unadjusted analyses, although chronic heart disease was associated with lower vaccination rates (S1 Table).

**Table 2. Vaccination prevalence according to population characteristics.**

| | Influenza Vaccination | | | Pneumococcal Vaccination | | |
|---|---|---|---|---|---|---|
| | Prevalence (%) | 95% CI | p-value‡ | Prevalence (%) | 95% CI | p-value‡ |
| **Overall** | 67.2 | 65.1–69.2 | | 20.8 | 18.7–23.0 | |
| Sex | | | 0.94 | | | 0.008 |
| Female | 67.1 | 63.9–70.1 | | 23.9 | 20.7–27.3 | |
| Male | 67.3 | 64.4–70.0 | | 18.1 | 15.6–21.1 | |
| Age groups | | | <0.001 | | | 0.47 |
| 70–79 years | 62.3 | 59.3–65.1 | | 20.3 | 17.6–23.4 | |
| 80–89 years | 72.1 | 68.7–75.1 | | 20.7 | 17.5–24.4 | |
| ≥90 years | 77.8 | 70.2–83.9 | | 24.3 | 16.9–33.5 | |
| Education | | | 0.011 | | | <0.001 |
| Less than high school | 71.3 | 67.5–74.9 | | 29.6 | 25.4–34.2 | |
| High school | 64.0 | 60.8–67.1 | | 13.6 | 11.1–16.6 | |
| University | 68.6 | 64.2–72.7 | | 22.7 | 18.8–27.3 | |
| Current smoker | | | 0.005 | | | 0.42 |
| Yes | 56.9 | 48.9–64.5 | | 17.8 | 11.6–26.2 | |
| No | 68.0 | 65.8–70.1 | | 21.1 | 18.9–23.4 | |
| Alcohol ≥ 12 SD per week | 65.9 | 58.8–72.4 | 0.69 | 18.4 | 12.3–26.7 | 0.49 |
| **Health care contacts*** | | | | | | |
| GP visits, n | | | <0.001 | | | <0.001 |
| 0 | 58.4 | 49.9–66.4 | | 7.2 | 3.3–15.2 | |
| 1–2 | 64.3 | 60.2–68.2 | | 17.1 | 13.7–21.2 | |
| 3–4 | 65.4 | 61.1–69.5 | | 20.9 | 16.9–25.5 | |
| ≥ 5 | 72.2 | 68.9–75.2 | | 25.4 | 22.0–29.1 | |
| Other outpatient physician or ED visits, n | | | 0.024 | | | 0.069 |
| 0 | 62.7 | 58.9–66.3 | | 19.3 | 16.1–22.9 | |
| 1–2 | 70.0 | 66.6–73.3 | | 19.6 | 16.5–23.2 | |
| ≥ 3 | 68.5 | 64.5–72.2 | | 24.9 | 20.5–30.0 | |
| Hospitalizations, n | | | 0.71 | | | 0.95 |
| 0 | 67.0 | 63.9–70.0 | | 21.2 | 18.3–24.4 | |
| 1 | 69.5 | 65.5–73.2 | | 19.6 | 16.0–23.7 | |
| ≥ 2 | 65.2 | 60.7–69.5 | | 21.9 | 17.5–26.9 | |
| Nursing home residents | 70.5 | 63.5–76.7 | 0.31 | 21.2 | 15.1–28.8 | 0.90 |
| Any home nursing visits | 68.8 | 64.6–72.7 | 0.36 | 23.1 | 19.0–27.9 | 0.22 |
| Receipt of informal care† | 73.6 | 69.2–77.6 | 0.001 | 27.1 | 22.1–32.7 | 0.005 |
| **Health Indexes** | | | | | | |
| EQ-5D < mean§ | 67.9 | 64.8–70.9 | 0.50 | 23.2 | 20.2–26.6 | 0.028 |
| CCI ≥ 7** | 72.8 | 69.3–76.0 | <0.001 | 21.8 | 18.2–25.8 | 0.51 |
| **Clinical risk groups** | | | | | | |
| Chronic heart disease | 68.1 | 65.2–70.9 | 0.34 | 18.7 | 16.0–21.7 | 0.044 |
| Chronic respiratory disease | 75.9 | 71.9–79.4 | <0.001 | 34.4 | 29.3–40.0 | <0.001 |
| Chronic liver disease | 60.4 | 51.0–69.1 | 0.12 | 19.8 | 13.0–29.0 | 0.81 |
| Chronic kidney disease | 77.3 | 69.9–83.4 | 0.006 | 23.0 | 16.1–31.7 | 0.54 |
| Diabetes mellitus | 68.7 | 65.0–72.2 | 0.29 | 23.7 | 20.0–27.8 | 0.06 |
| Rheumatic disease | 68.1 | 60.5–74.9 | 0.78 | 24.6 | 17.5–33.3 | 0.29 |
| Any malignancy†† | 67.9 | 63.7–71.9 | 0.66 | 22.8 | 18.4–27.8 | 0.33 |

(*Continued*)

**Table 2.** (Continued)

|  | Influenza Vaccination | | | Pneumococcal Vaccination | | |
|---|---|---|---|---|---|---|
|  | Prevalence (%) | 95% CI | p-value‡ | Prevalence (%) | 95% CI | p-value‡ |
| Immunosuppression | 65.4 | 57.8–72.4 | 0.63 | 27.4 | 20.0–36.4 | 0.07 |

Unadjusted vaccination prevalence and its 95% CI is presented by subgroups. Abbreviations: CCI, Charlson comorbidity index; CI, confidence interval; SD, standard drinks; ER, emergency room; GP, general practitioner.

‡ In case of age, GP visits, other outpatient physician or ED visits, and hospitalizations, the p-value refers to a p for trend.

* Health care contacts refer to hospitalizations within 12 months, or GP visits, ED or outpatient clinic/specialist visits, receipt of informal care, any nursing home visits, or permanent nursing home residency within 6 months prior to the baseline visit.

† defined as care received by relatives or other close persons.

§ Questionnaire-based health status on a 1 to 0 scale. A value of 1 corresponds to perfect health and a value of 0 to death.

** The CCI predicts 10-year survival in patients with multiple comorbidities and ranges from 0 to 33 points. Lower scores indicate a higher risk of 10-year-survival. 7 points correspond to an estimated 0% 10-year survival.

†† Except malignant neoplasm of skin.

In a sensitivity analysis excluding all 453 patients from Switzerland, chronic respiratory disease, diabetes, immunosuppression and GP visits remained independently associated with higher pneumococcal vaccination rates, as was chronic liver disease (PR 1.04, 95% CI 1.01.-1.06) as well as a higher CCI (PR 1.25, 95% CI 1.15–1.37 for CCI ≥7 vs. CCI <7, S3 Table).

## Discussion

In this population of multimorbid older inpatients aged ≥70 years with polypharmacy, 67% had received an influenza vaccination in the last 12 months, but only one in five persons had received a pneumococcal vaccination. Although patients with chronic respiratory disease and diabetes were more likely to be vaccinated, vaccination uptake in high risk groups remained insufficient. Outpatient medical contacts, particularly GP visits, were associated with a higher vaccination prevalence.

All participating countries recommend the influenza vaccination in all individuals aged 65 years or older [4]. The prevalence of influenza vaccination in our study was consistent with one cross-sectional study of hospitalized older patients in Spain and with estimates from the annual epidemiological survey by the European Centre for Disease Prevention and Control (ECDC) [23, 42]. According to the ECDC, the average influenza vaccination rates in the influenza season 2017–2018 were 47.1% in older age groups and 44.9% in individuals with chronic medical conditions; influenza vaccination rates in central European countries were mostly higher than 50% and in the UK over 60% [18]. The ECDC's goal of 75% influenza vaccination rate in people above 65 years of age by 2015 has not been reached by any country in Europe [18], but some specific subgroups in our study have indeed vaccination uptakes near and over the ECDC's goal. Specifically, we found high influenza vaccination coverage >75% among persons over 90 years (77.8%) as well as in those with chronic respiratory diseases (75.9%) and chronic kidney diseases (77.3%).

Overall, the prevalence of pneumococcal vaccination in our population of multimorbid older inpatients was very low with 20.8% and remained low in high risk population including those with chronic respiratory disease, immunosuppression, or chronic heart disease (all <35%). Pneumococcal vaccination coverage among older adults in European countries varies between 10% and 69% [22, 42, 43], although the available data is very limited [22]. The US office of disease prevention and health promotion set the Healthy People 2020 goal for pneumococcal vaccination uptake at 60% for at-risk adults aged <65 years and 90% for older people

**Table 3. Association of chronic health conditions and medical resource utilization with receipt of influenza and pneumococcal vaccination.**

| | Adjusted PR§ | 95% CI | p-value‡ | Adjusted PR§ | 95% CI | p-value‡ |
|---|---|---|---|---|---|---|
| | Influenza Vaccination | | | Pneumococcal Vaccination | | |
| **Clinical risk groups** | | | | | | |
| Chronic heart disease | 0.99 | 0.96–1.03 | 0.71 | 0.81 | 0.63–1.03 | 0.09 |
| Chronic respiratory disease | 1.09 | 1.03–1.16 | 0.003 | 2.03 | 1.22–3.40 | 0.007 |
| Chronic liver disease | 0.94 | 0.79–1.13 | 0.53 | 0.98 | 0.62–1.55 | 0.93 |
| Chronic kidney disease | 1.12 | 1.08–1.17 | <0.001 | 1.07 | 0.81–1.42 | 0.62 |
| Diabetes mellitus | 1.06 | 1.03–1.08 | <0.001 | 1.24 | 1.16–1.34 | <0.001 |
| Rheumatic disease | 1.01 | 0.91–1.12 | 0.84 | 1.10 | 0.83–1.47 | 0.51 |
| Any malignancy† | 1.04 | 0.99–1.09 | 0.09 | 1.18 | 0.92–1.50 | 0.19 |
| Immunosuppression | 0.97 | 0.85–1.12 | 0.70 | 1.29 | 1.03–1.61 | 0.028 |
| **Health care contacts*** | | | | | | |
| GP visits, n | | | | | | |
| 0 | Reference | | | Reference | | |
| 1–2 | 1.08 | 0.86–1.37 | 0.16 | 2.29 | 1.59–3.31 | <0.001 |
| 3–4 | 1.08 | 0.92–1.28 | | 2.88 | 2.75–3.02 | |
| ≥ 5 | 1.21 | 0.91–1.61 | | 3.41 | 2.74–4.24 | |
| Other outpatient physician or ED visits, n | | | | | | |
| 0 | Reference | | | Reference | | |
| 1–2 | 1.12 | 1.02–1.22 | 0.027 | 1.07 | 0.64–1.78 | 0.47 |
| ≥3 | 1.12 | 1.01–1.24 | | 1.42 | 0.54–3.72 | |
| Hospitalizations, n | | | | | | |
| 0 | Reference | | | Reference | | |
| 1 | 1.05 | 0.98–1.12 | 0.67 | 0.92 | 0.82–1.03 | 0.88 |
| ≥ 2 | 0.98 | 0.92–1.06 | | 1.02 | 0.80–1.30 | |
| Nursing home resident | 1.03 | 0.93–1.14 | 0.56 | 0.99 | 0.65–1.50 | 0.96 |
| Any home nursing visits | 1.00 | 0.97–1.04 | 0.83 | 1.13 | 0.90–1.42 | 0.30 |
| Receipt of informal care†† | 1.19 | 0.97–1.29 | 0.11 | 1.34 | 0.73–2.44 | 0.34 |
| **Health scores** | | | | | | |
| EQ-5D < mean§§ | 1.04 | 0.99–1.09 | 0.13 | 1.27 | 0.71–2.28 | 0.43 |
| CCI ≥ 7** | 1.10 | 1.02–1.19 | 0.015 | 1.05 | 0.91–1.21 | 0.52 |

Abbreviations: CCI, Charlson comorbidity index; CI, confidence interval; ED, emergency room; GP, general practitioner; PR, prevalence ratio.

§ adjusted for age, sex, ethnicity, education, alcohol consumption and smoking status.

‡ In case of GP visits, other outpatient physician or ED visits, and hospitalizations is the p-value refers to a p for trend.

† Except malignant neoplasm of skin.

* Health care contacts refer to hospitalizations within 12 months, or GP visits, ED or outpatient clinic/specialist visits, receipt of informal care, any nursing home visits, or permanent nursing home residency within 6 months prior to the baseline visit.

†† defined as care received by relatives or other close persons.

§§ Questionnaire-based health status on a 1 to 0 scale. A value of 1 corresponds to perfect health and a value of 0 to death.

** The CCI predicts 10-year survival in patients with multiple comorbidities and ranges from 0 to 33 points. Lower scores indicate a higher risk 10-year-survival. 7 points correspond to an estimated 0% 10-year survival.

[44]; thus, the pneumococcal vaccination rates as found in our study and in other European countries are far off this goal [22, 42, 43].

Principal sociodemographic determinants of influenza and pneumococcal vaccination uptake in our study were older age, lower education, and non-smoking-status. These factors seem to positively influence vaccination uptake across different countries and cultures [19, 42, 43, 45, 46]. Although smoking is related to a wide range of chronic diseases it has been

associated with lower influenza and pneumococcal vaccination rates in a large survey among community-dwelling older individuals in the United States [47]. These are at increased risk to suffer from the vaccine-preventable diseases under study and, consequently, vaccination rates should be higher in these groups [48, 49]. Further investigation is needed to unveil the reasons for not vaccinating, and efforts are necessary for promotion and education concerning vaccinations in this vulnerable group. Our results showed a U-shaped relationship between education and prevalence of influenza or pneumococcal vaccination. Previous studies have found inconsistent results between education and vaccination uptake, with some studies showing highest vaccination prevalence in individuals with low education and others in those with higher education [23, 45, 46]. Overall, the association is complex and differs according to population, culture, and health care system, and underlying factors merit further research.

An important factor influencing vaccination rates is chronic illness [46, 50]. Chronic respiratory disease showed the strongest association with vaccination rates in our and other studies [46, 50]. Both influenza and pneumococcal disease affect the lungs and are major triggers for exacerbations of asthma and chronic obstructive pulmonary disease (COPD) [51]. Therefore, it is probable that treating physicians recommend the vaccination especially to this vulnerable population. Other chronic illnesses associated with increased vaccination uptake were chronic kidney disease and diabetes mellitus, as previously observed in Switzerland [46]. Conversely, chronic heart disease showed no association with vaccination rates in our study, despite their increased risk to develop pneumococcal disease and its complications [7, 52] and the reduction of cardiovascular events with influenza vaccination in this group [53]. It has been previously observed that influenza vaccination rates are suboptimal in those with chronic heart disease [54], but not uniformly so. Recently published results from a population-based observational study from Catalonia showing a strong positive association between chronic heart disease and pneumococcal vaccination rates. This different result could be due to a stronger promotion of pneumococcal vaccination in Spain compared to other European countries [55]. Overall awareness should be raised for this vulnerable population. Similarly, we didn't find any association of other high-risk group status with increased vaccination rates, like in those with chronic liver disease, rheumatic disease or malignant disease. Thus, the need in educating doctors about the indications and effectiveness of these vaccinations in high risk groups is still large, in order to prevent unnecessary and potentially preventable disease.

One of the most important factors associated with vaccination rates was the number and type of health care contacts. In line with previous results in hospitalized older patients [23], those with a higher number of GP and other outpatient medical visits were more likely to be vaccinated in our study. Previous studies have shown that the vaccination recommendations from medical staff, especially from GP's, are among the strongest predictors for actual vaccination and it is the GP, who performs 93% of all vaccinations [20, 43, 50, 56, 57]. However, given the fact that a third of hospitalized multimorbid older patients are not up to date with their influenza vaccine and more than 75% of all high risk patients have not received the pneumococcal vaccinations, efforts are urgently needed to further educate physicians and to increase vaccination rates, particularly targeting patients with few outpatient physician contacts. This could be achieved by extensive patient and community-wide education for example by sending informational letters, as previously demonstrated in a Californian randomized trial [58], or provision of vaccination services at pharmacies. In addition, hospitalization for acute illness could be another opportunity to educate patients, and in the absence of acute contraindications, to vaccinate them, as vaccination is safe and effective in hospitalized patients and increases vaccination rates substantially as shown previously [59–61].

## Strengths and limitations

To the best of our knowledge this is the first study investigating influenza and pneumococcal vaccination rates and determinants for these vaccinations in hospitalized multimorbid, older people across different European countries. The main strength of our study is our large and high-quality data on comorbidities and the use of medical resources. In addition, it addresses older and multimorbid patients, a population that is underrepresented in clinical research.

Our study has some limitations. Hospitalized patients were included in this study, therefore these results might not translate to the general population. Vaccination information was self-reported, as it was in most other studies reporting influenza or pneumococccal vaccination rates [20, 50, 57]. Self-reporting has been shown to be a highly accurate method of assessing influenza vaccination status in older patients, but moderately accurate for pneumococcal vaccination, which is administered only once [62]. The prevalence of pneumococcal vaccination is underestimated by 5–10% when self-reported, as shown previously [63]. We cannot exclude that study power has been insufficient to detect statistically significant associations between influenza vaccination and numbers of GP visits, as well as pneumococcal vaccination and other outpatient physician / ED visits, as only non-statistically significant trends have been observed for these associations. Public financial support and reimbursement of vaccinations differ between countries (particularly for pneumococcal vaccination), which may have affected our results. Furthermore, we did not measure the number of pharmacy contacts as a potential determinant for receipt of vaccinations, given that they also provide these vaccinations. Another limitation is the cross-sectional design, which impedes any conclusion about the temporal relationship of patient characteristics and vaccination status.

## Conclusion

Influenza and pneumococcal vaccination are determined by comorbidities as well as number and type of health care contacts, but uptake of these two vaccinations remains insufficient in this population of European multimorbid older inpatients. Our study showed that an increasing number of GP visits was strongly associated with a higher prevalence of pneumococcal vaccination, underlining the important role of the GP in the provision of recommended vaccinations. Further efforts are urgently needed to increase vaccination rates in these patients who are at particular high risk of complications from these potentially preventable diseases. Future interventions to promote appropriate vaccinations are needed, and hospitalization for acute illness could be taken as an opportunity to promote vaccination, particularly targeting patients with few outpatient physician contacts.

## Supporting information

**S1 Table. Association of chronic health conditions and medical resource utilization with receipt of influenza and pneumococcal vaccination.**
(DOCX)

**S2 Table. Association of chronic health conditions and medical resource utilization with receipt of influenza and pneumococcal vaccination, calculated with Poisson regression model.**
(DOCX)

**S3 Table. Association of chronic health conditions and medical resource utilization with receipt of pneumococcal vaccination excluding Switzerland.**
(DOCX)

## Author Contributions

**Conceptualization:** Dimitrios David Papazoglou, Oliver Baretella, Martin Feller, Elisavet Moutzouri, Drahomir Aujesky, Denis O'Mahony, Wilma Knol, Olivia Dalleur, Nicolas Rodondi, Christine Baumgartner.

**Data curation:** Dimitrios David Papazoglou, Oliver Baretella.

**Formal analysis:** Dimitrios David Papazoglou, Cinzia Del Giovane.

**Investigation:** Oliver Baretella.

**Methodology:** Dimitrios David Papazoglou, Martin Feller, Cinzia Del Giovane, Elisavet Moutzouri, Drahomir Aujesky, Matthias Schwenkglenks, Christine Baumgartner.

**Supervision:** Cinzia Del Giovane, Denis O'Mahony, Wilma Knol, Olivia Dalleur, Nicolas Rodondi, Christine Baumgartner.

**Validation:** Cinzia Del Giovane.

**Visualization:** Dimitrios David Papazoglou, Christine Baumgartner.

**Writing – original draft:** Dimitrios David Papazoglou, Christine Baumgartner.

**Writing – review & editing:** Dimitrios David Papazoglou, Oliver Baretella, Martin Feller, Cinzia Del Giovane, Elisavet Moutzouri, Drahomir Aujesky, Matthias Schwenkglenks, Denis O'Mahony, Wilma Knol, Olivia Dalleur, Nicolas Rodondi, Christine Baumgartner.

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
