## [Decision Letter · Decision Letter 0]

16 Aug 2021

PONE-D-21-21995

Cross-sectional study on the prevalence of influenza and pneumococcal vaccination and its association with health conditions and risk factors among hospitalized multimorbid older patients

PLOS ONE

Dear Dr. Papazoglou,

Thank you for submitting your manuscript to PLOS ONE. After careful consideration, we feel that it has merit but does not fully meet PLOS ONE’s publication criteria as it currently stands. This is an interesting study. Its design is appropriate and the manuscript is well organized, clear and easy to read.   However, there are several comments of the reviewers to be addressed. Therefore, we invite you to submit a revised version of the manuscript that responds to the points raised during the review process.

We look forward to receiving your revised manuscript.

Kind regards,

Juan F. Orueta, MD, PhD

Academic Editor

PLOS ONE

Journal Requirements:

2. In the ethics statement in the Methods and online submission information, please specify the type of informed consent that was obtained from the participants (for instance, written or verbal, and if verbal, how it was documented and witnessed).

The registration number of the OPERAM trial should be included in the Methods. 

Also, please amend your current ethics statement to include the full name of the ethics committee/institutional

review boards that approved your specific study.

Additional Editor Comments:

I have noticed in table 1 that some values are missing in a few patients. For example, in the column ”Influenza Vaccination Yes” adding up the 3 education groups the total is 1,303 patients (instead of 1,317) or the total for the 4 GP visits groups is 1,300. Such situation is usual in large databases, but it should be explained in the manuscript to avoid misunderstandings. Are there also missing values for other variables (smoking status or others)?

Reviewers' comments:

Reviewer's Responses to Questions

**Comments to the Author**

1. Is the manuscript technically sound, and do the data support the conclusions?

Reviewer #1: Yes

Reviewer #2: Partly

Reviewer #3: Yes

2. Has the statistical analysis been performed appropriately and rigorously? 

Reviewer #1: Yes

Reviewer #2: No

Reviewer #3: Yes

3. Have the authors made all data underlying the findings in their manuscript fully available?

Reviewer #1: No

Reviewer #2: No

Reviewer #3: Yes

4. Is the manuscript presented in an intelligible fashion and written in standard English?

Reviewer #1: Yes

Reviewer #2: Yes

Reviewer #3: Yes

5. Review Comments to the Author

Reviewer #1: 1) “Furthermore, we used clustered analyses by study sites to take into account the participants' correlation within each site”—it is unclear what exactly was done? What’s more, the relevant results was only briefly mentioned in the Discussion section. Shouldn’t include the site as a covariate in the primary models?

2) “Those who received an influenza vaccination tended to have fewer years of formal schooling”—this is an inaccurate and exaggerated statement on the results since for those with the University level education, the vaccination rate increased.

Reviewer #2: Main comment:

The manuscript “Cross-sectional study on the prevalence of influenza and pneumococcal vaccination and its association with health conditions and risk factors among hospitalized multimorbid older patients” has been reviewed. The topic of the study is of interest because despite the fact that influenza and pneumococcal vaccination is recommended for people aged 65 and older, the coverage is suboptimal and, therefore, knowledge about factors associated with vaccination in this population is needed. However, the authors only report the results of multivariate analyses and results of bivariate analyses should be shown to the readers.

Specific comments:

Page 2, line 41: “The prevalence remained low” should be changed to “The prevalence of vaccination remained low”

Page 2, line 45: “95% confidence Interval [C] I…” should be changed to “95% confidence Interval [CI]…”

Page 2. Line 3: “Streptococcus pneumonia” should be changed to “Streptococcus pneumoniae”

Statistical methods and Results: Only Adjusted OR are shown in table 3. Authors should report unadjusted and adjusted OR.

In table 2, the unadjusted PR and 95% CI should be included

Page 11, line 244: “with prevalence ratios” should be changed to “with adjusted prevalence ratios”.

Discussion is too long and should be shortened.

In references 8, 11, 12, 16, 19, 22, 23, 24, 29, 31, 33, 34, 35, 39, 45, 55 and 57 the capital letters of all the words in the title of the article (with the exception of the first word) should be changed to small letters.

Reviewer #3: The authors tried to identify determinants of inappropriate lack of vaccination in the elderly with multimorbid and taking five or more chronic medications. The topic is crucial to reduce preventable diseases in this vulnerable population. Nevertheless, the reviewer thinks that several minor concerns remain in this current manuscript as follows.

Minor points

The authors did not mention why they focus on patients older than 70 with multimorbid (3 or more) and taking multi-medicine (5 or more). Older adults are generally defined as 65 or older. Please, mention any reasons for focusing on the targeted population in the study.

For both influenza and pneumococcal vaccine, the public subsidy might affect vaccination rates. The system of public financial support might vary across the countries. If the authors have such information, the factor needs to be considered.

For both influenza and pneumococcal vaccine, education is significantly associated with vaccination behavior (Table 2). However, U-shaped prevalences are observed in both, which means prevalences are higher in low education (less than high school) and high education (university) than in the middle (high school). Please, discuss some reason for this.

In multivariable analysis (Table 3), a significant association with a specialized physician or ED visit (not GP visit) was found in influenza. On the other, such association with GP visits (not specialized physician or ED visit) was observed in pneumococcal vaccination. Please, discuss this difference and its reason.

Please, mention the reason for choosing log-binomial regression models.

6. PLOS authors have the option to publish the peer review history of their article (what does this mean?). If published, this will include your full peer review and any attached files.

Reviewer #1: No

Reviewer #2: No

Reviewer #3: **Yes: **Yugo Shobugawa

---

## [Author Response · Author response to Decision Letter 0]

29 Sep 2021

Dimitrios Papazoglou

Institute of Primary Health Care (BIHAM)

University of Bern

Mittelstrasse 43

CH-3012 Bern

Switzerland

e-mail: papazoglou2019@gmail.com

phone: +41 (0)797576013

 Juan F. Orueta, MD, PhD

 Academic Editor

 PLOS ONE

 Bern, Switzerland, 09/29/2021

Re: Revision of PONE-D-21-21995 “Cross-sectional study on the prevalence of influenza and pneumococcal vaccination and its association with health conditions and risk factors among multimorbid older patients”

Dear Dr. Orueta

We thank you for the constructive review of this manuscript and for inviting us to submit a revised version to PLOS ONE. We have carefully considered your comments and have revised the manuscript accordingly. All modifications are outlined below and highlighted in the manuscript. 

In the funding information the last sentence was added to match the statement in the publication of the main trial [1]: “The funder of the study had no role in study design, data collection, data analysis, data interpretation or writing of the report.” Otherwise, there was no change in the funding information or financial disclosures.

This study involves human research participant data containing sensitive patient information. In the EU Horizon 2020 grant agreement for the OPERAM study, it had been specified that the data will be made available upon request if the use has been approved by an ethical committee. Therefore, restrictions to make the underlying data directly publicly available are both due to legal and ethical reasons, as health data are sensitive data. 

Data for this study will be made available for scientific purposes upon request for researchers whose proposed use of the data has been approved by the OPERAM publication committee. After approval and signing of a data transfer agreement ensuring adherence to privacy and data handling, data and documentation will be made available through a secure file exchange platform. Partially deidentified participant data, a data dictionary and annotated case report forms will be made available. For data access, external researchers can contact operam@biham.unibe.ch [1].

Each author has approved the revised version. We hope that these changes meet with your approval. Please direct further questions or comments to the corresponding author Dimitros Papazoglou (papazoglou2019@gmail.com).

Sincerely,

Dimitrios Papazoglou (corresponding author, contact information above) 

Journal Requirements:

We have adjusted the manuscript accordingly.

2. In the ethics statement in the Methods and online submission information, please specify the type of informed consent that was obtained from the participants (for instance, written or verbal, and if verbal, how it was documented and witnessed).

The registration number of the OPERAM trial should be included in the Methods. 

Also, please amend your current ethics statement to include the full name of the ethics committee/institutional review boards that approved your specific study.

We have added the registration number of the OPERAM trial on page 5, line 99: “ClinicalTrials.gov Identifier: NCT02986425”.

The ethics statement was amended accordingly on page 5, lines 111-116: 

“The local ethics committee at each of the participating sites approved the protocol of the OPERAM study (Cantonal Ethics Committee Bern, Switzerland; Medical Research Ethics Committee Utrecht, Netherlands; Comité d’Ethique Hospitalo-Facultaire Saint-Luc-UCL, Belgium; Cork University Teaching Hospitals Clinical Ethics Committee, Ireland). All participants or their legal representatives provided written informed consent.”

When you resubmit, please ensure that you provide the correct grant numbers for the awards you received for your study in the ‘Funding Information’ section. We adapted the grant information in the “Funding information” and “Financial Disclosure” section and included the updated statement in the cover letter as follows: 

“This work is part of the project “OPERAM: OPtimising thERapy to prevent Avoidable hospital admissions in the Multimorbid elderly” supported by the European Union's Horizon 2020 research and innovation program under the grant agreement No 634238, and by the Swiss State Secretariat for Education, Research and Innovation (SERI) under contract number 15.0137. The opinions expressed and arguments employed herein are those of the authors and do not necessarily reflect the official views of the European Commission and the Swiss government. This project was also partially funded by the Swiss National Scientific Foundation (SNSF 320030_188549). The funder of the study had no role in study design, data collection, data analysis, data interpretation or writing of the report.”

We now also added the funding statement to the manuscript on page 18/19. 

This study involves human research participant data containing sensitive patient information. In the EU Horizon 2020 grant agreement for the OPERAM study, it was specified that the data will be made available upon request if the use has been approved by an ethical committee. Therefore, restrictions to make the underlying data directly publicly available are both due to legal and ethical reasons, as health data are sensitive data. 

Data for this study will be made available for scientific purposes upon request for researchers whose proposed use of the data has been approved by the OPERAM publication committee. After approval and signing of a data transfer agreement ensuring adherence to privacy and data handling, data and documentation will be made available through a secure file exchange platform. Partially deidentified participant data, a data dictionary and annotated case report forms will be made available. For data access, external researchers can contact operam@biham.unibe.ch[1].

We have now added this information to the cover letter. 

Please see our answer to your comment 4a above.

We have reviewed the reference list. The following change was made: The electronic reference from the WHO about pneumococcal disease was not available any more [2], so we added two new appropriate citations for the corresponding statement in the manuscript (our new references 5 and 6 in the manuscript on page 3, lines 68 and 69) [3, 4].

Additional Editor Comments:

1. I have noticed in table 1 that some values are missing in a few patients. For example, in the column ”Influenza Vaccination Yes” adding up the 3 education groups the total is 1,303 patients (instead of 1,317) or the total for the 4 GP visits groups is 1,300. Such situation is usual in large databases, but it should be explained in the manuscript to avoid misunderstandings. Are there also missing values for other variables (smoking status or others)?

We have now indicated the number of missing values for each of the variables in the footnote of table 1:

“The number of missing data in participants included in the analysis on influenza / pneumococcal vaccination, respectively, was: education n=29 / n=27, ethnicity n=6 / n=5, smoking n=10 / n=8, alcohol n=17 / n=15, GP visits n=18 / n=15, other outpatient physician or ED visits n=17 / n=15, hospitalization n=13 / n=10, nursing home resident n=10 / n=8, any home nursing visits n=16 / n=14, receipt of informal care n=15 / n=13, and n=4 /n=3 for all of the chronic health conditions defining the clinical risk groups.”

Reviewer #1: Comments to the Author

1. “Furthermore, we used clustered analyses by study sites to take into account the participants' correlation within each site”—it is unclear what exactly was done? What’s more, the relevant results was only briefly mentioned in the Discussion section. Shouldn’t include the site as a covariate in the primary models?

We thank the reviewer for this comment. For this analysis, we used data from the multicenter OPERAM trial, which was conducted at 4 sites from 4 different countries. Given that the population, health care systems, and management practices may slightly differ between the countries, participants from one site are more similar to each other than to participants from other sites. To take this into account, we clustered all our regression analyses by study site. Including site as a covariate would have only adjusted for it, but clustering would not have been taken into account. We clarified this in the Methods on page 7, lines 175-177: 

“Furthermore, we clustered the analyses by study sites to take into account the participants’ correlation within each site, thus allowing for intragroup correlation of standard errors.”

2. “Those who received an influenza vaccination tended to have fewer years of formal schooling”—this is an inaccurate and exaggerated statement on the results since for those with the University level education, the vaccination rate increased.

We agree with the reviewer and replaced the sentence in the Results section accordingly (see page 10, lines 234 and 237):

“The prevalence of influenza vaccination was higher in individuals with low (less than high school, 71.3%) and high education levels (University, 68.6%) compared to those with an intermediate education level (high school, 64.0%).” 

Reviewer #2: Comments to the Author

1. Main comment:

The manuscript “Cross-sectional study on the prevalence of influenza and pneumococcal vaccination and its association with health conditions and risk factors among hospitalized multimorbid older patients” has been reviewed. The topic of the study is of interest because despite the fact that influenza and pneumococcal vaccination is recommended for people aged 65 and older, the coverage is suboptimal and, therefore, knowledge about factors associated with vaccination in this population is needed. However, the authors only report the results of multivariate analyses and results of bivariate analyses should be shown to the readers.

Thank you very much for your comment. Concerning the results of bivariate analyses, please see our answer to your comment 5.

Specific comments:

2. Page 2, line 41: “The prevalence remained low” should be changed to “The prevalence of vaccination remained low”

We changed the sentence as suggested (page 2, line 41-43): 

“The prevalence of vaccination remained low in high-risk populations with chronic respiratory disease (34%) or diabetes (24%), but increased with an increasing number of outpatient medical contacts.”

3. Page 2, line 45: “95% confidence Interval [C] I…” should be changed to “95% confidence Interval [CI]…”

Thank you, we changed this typo accordingly.

4. Page 2. Line 3: “Streptococcus pneumonia” should be changed to “Streptococcus pneumoniae”

We adapted this accordingly on page 3 line 66: 

“Manifestations of pneumococcal disease caused by infection with Streptococcus pneumoniae include community-acquired pneumonia, meningitis, as well as severe invasive pneumococcal disease.”

5. Statistical methods and Results: Only Adjusted OR are shown in table 3. Authors should report unadjusted and adjusted OR.

We report the adjusted prevalence ratios in table 3 and the unadjusted prevalence ratios in the supplemental table 1 (S1 Table), as also indicated in the Results section on page 11, line 259, and page 12 line 286. By mistake, we had listed adjustment variables in the footnotes of S1 Table (symbol § next to PR on the first line), which may have resulted in this confusion. We have now corrected the mistake and removed this footnote, as the results shown in S1 Table refer to the unadjusted results. The unadjusted analyses correspond to the bivariate analyses, as we used each variable at a time. 

6. In table 2, the unadjusted PR and 95% CI should be included

In table 2 we report on the vaccination prevalence with the corresponding 95% confidence interval by different subgroups; these results do not refer to prevalence ratios, and are all unadjusted. We have now clarified this in the footnote of Table 2:

“Crude vaccination prevalence and its 95% CI is presented by subgroups." 

As mentioned in our answer to your comment 5, there was a mistake in the footnote of supplemental table 1 (S1), which may have lead to the confusion.

7. Page 11, line 244: “with prevalence ratios” should be changed to “with adjusted prevalence ratios”.

We changed the sentence as suggested (page 11, line 259-263): 

“In multivariable analysis, presence of chronic respiratory disease, chronic kidney disease and diabetes mellitus were independently associated with a higher prevalence of influenza vaccination, with adjusted prevalence ratios (PR) of 1.09 (95% CI 1.03-1.16), 1.12 (95% CI 1.08-1.17), and 1.06 (95% CI 1.03-1.08), respectively, while immunosuppression showed no association (Table 3).”

8. Discussion is too long and should be shortened.

We revised and shortened the discussion section, condensing the information provided (see page 15, lines 344-347 and lines 366-367; page 16, lines 371-377 and 385-391; and page 17, lines 397-398).

9. In references 8, 11, 12, 16, 19, 22, 23, 24, 29, 31, 33, 34, 35, 39, 45, 55 and 57 the capital letters of all the words in the title of the article (with the exception of the first word) should be changed to small letters.

We thank the reviewer for the comment and changed the abovementioned references accordingly.

Reviewer #3: Comments to the Author

The authors tried to identify determinants of inappropriate lack of vaccination in the elderly with multimorbid and taking five or more chronic medications. The topic is crucial to reduce preventable diseases in this vulnerable population. Nevertheless, the reviewer thinks that several minor concerns remain in this current manuscript as follows.

Minor points

1. The authors did not mention why they focus on patients older than 70 with multimorbid (3 or more) and taking multi-medicine (5 or more). Older adults are generally defined as 65 or older. Please, mention any reasons for focusing on the targeted population in the study.

For this cross-sectional study we made use of the OPERAM trial database, and the inclusion criteria for the OPERAM trial were multimorbidity (defined by ≥ 3 chronic health conditions), age ≥70 years, and polypharmacy (defined ≥ 5 chronic medications). While this is a population which is underrepresented in most clinical trials, these characteristics are very prevalent among hospitalized patients [5-7]. Overall, there is a lack of vaccination data on these particularly vulnerable patients in Europe. We clarified that the inclusion criteria for this study were based on the inclusion criteria of the OPERAM trial (page 5, line 105-107 and 109-110):

“To be eligible for the OPERAM trial, patients had to be ≥70 years of age, multimorbid (defined as having ≥3 chronic health conditions) and had to take five or more chronic medications. […] We considered all OPERAM participants with available baseline data on vaccination status for the current study.”

2. For both influenza and pneumococcal vaccine, the public subsidy might affect vaccination rates. The system of public financial support might vary across the countries. If the authors have such information, the factor needs to be considered.

We thank the reviewer for this important comment. For older individuals aged at least 65 years from Ireland, Switzerland, and the Netherlands, costs for the influenza vaccination are fully covered by the national health services or national health insurances [8, 9]. For participants recruited at the study site in Belgium (Louvain, located in the Flemish community), influenza vaccines are covered for residents in elderly homes and residents in other health care institutions, and may be partially reimbursed for persons aged 65 years or older if they have a prescription [8, 10]. Since vaccination costs are fully covered in three of the four countries, and at least partly in the fourth country from which participants were included in our study, this fact may not have substantially affected our results on influenza vaccination.

In case of the pneumococcal vaccination, the Netherlands and Belgium do not fund the costs for vaccination [11]. In Switzerland and Ireland, costs are fully covered through the national health insurance scheme and national health service, respectively [9, 11]. This difference may have had an impact on pneumococcal vaccination rates [12].

We have now added a statement accordingly to the limitations section on page 18, line 429-431:

“Public financial support and reimbursement of vaccinations differ between countries (particularly for pneumococcal vaccination), which may have affected our results.”

3. For both influenza and pneumococcal vaccine, education is significantly associated with vaccination behavior (Table 2). However, U-shaped prevalences are observed in both, which means prevalences are higher in low education (less than high school) and high education (university) than in the middle (high school). Please, discuss some reason for this.

We appreciate the reviewers’ comment on the association of education and vaccination prevalence. In the literature, a positive association of education and vaccination prevalence is well described [13, 14], although findings are not consistent and vary widely. For example, some studies from Western Europe did not show any association of educational level and influenza or pneumococcal vaccination uptake [15, 16], while others have found a higher influenza vaccination coverage in individuals with lower education [17, 18].

Factors influencing vaccination coverage are multifaceted. Higher education is associated with higher socio-economic status, which is correlated with better health insurance coverage and better access to health care providers, two major reasons for vaccination uptake [19, 20]. A population with higher education is probably also better informed about vaccinations, which has been shown to be another important factor for vaccination uptake [21].

Lower socio-economic status is associated with lower health status [22]. This could lead to increased health care contacts, which has been shown to be one of the strongest predictors of vaccination uptake in our as well as in other studies [14, 21, 23]. Another possible explanation for higher vaccination rates in individuals with lower education may be the fact that negative attitudes towards vaccination may be less frequent [24], and that these persons are less likely to construct a narrative supporting their preconceptions by searching for information on the internet or other sources [25, 26]. 

Overall, the association between education and vaccination prevalence is complex and differs according to population, culture, and health care system [13-21, 23]. We have now commented on this finding in the manuscript accordingly on page 15, line 357-363: 

“Our results showed a U-shaped relationship between education and prevalence of influenza or pneumococcal vaccination. Previous studies have found inconsistent results between education and vaccination uptake, with some studies showing highest vaccination prevalences in individuals with low education and others in those with higher education [13, 15, 18]. Overall, the association is complex and differs according to population, culture, and health care system, and underlying factors merit further research.” 

4. In multivariable analysis (Table 3), a significant association with a specialized physician or ED visit (not GP visit) was found in influenza. On the other, such association with GP visits (not specialized physician or ED visit) was observed in pneumococcal vaccination. Please, discuss this difference and its reason.

We appreciate the reviewers’ comment. A strong association between health care provider contacts and influenza as well as with pneumococcal vaccination has been previously described [14, 21, 23, 27, 28]. General practitioner (GP) visits as well as other outpatient visits have been shown to be associated with higher vaccination uptake of both vaccinations [21, 28, 29]. Our results still show a non-statistically significant trend towards a higher vaccination prevalence with more GP visits for influenza and other outpatient physician visits / ED visits for pneumococcal vaccination (see our Table 3). 

We cannot exclude that insufficient power may explain the lack of a significant association of influenza vaccination with GP visits as well as pneumococcal vaccination with specialist/ED visits. In addition, general practitioners may be more likely to discuss the pneumococcal vaccination with the patients compared to specialists or ED physicians, because it is a one-time vaccination: a previous study in Germany found that 93% of all pneumococcal vaccinations are administered by the general practitioner [23]. On the other hand, specialists or ED physicians may be more alert to providing the yearly influenza vaccinations.

We have now added a sentence to the limitation section accordingly (page 18, lines 426-431):

“We cannot exclude that study power has been insufficient to detect statistically significant associations between influenza vaccination and numbers of GP visits, as well as pneumococcal vaccinations and other outpatient physician / ED visits, as only non-statistically significant trends have been observed for these associations.”

5. Please, mention the reason for choosing log-binomial regression models.

The outcome of our study is a prevalence which is why we calculated prevalence ratios. It has been shown that prevalence ratios are epidemiologically und clinically highly interpretable and significant [30-32]. The log-binomial regression models are considered the most appropriate models to estimate prevalence ratios as previously suggested [32, 33]. For clarification the following sentence was added (page 7, lines 170-172):

“We used log-binomial regression models because this is the recommended method to estimate prevalence ratios [32, 33].” 

References

1. Blum MR, Sallevelt BTGM, Spinewine A, O’Mahony D, Moutzouri E, Feller M, et al. Optimizing Therapy to Prevent Avoidable Hospital Admissions in Multimorbid Older Adults (OPERAM): cluster randomised controlled trial. BMJ. 2021;374:n1585.

2. World Health Organization. Pneumococcal disease. 2019. Available from: https://www.who.int/ith/diseases/pneumococcal/en/.

3. UNAIDS. Global HIV & AIDS statistics — Fact sheet. 2021. Available from: https://www.unaids.org/en/resources/fact-sheet.

4. Estimates of the global, regional, and national morbidity, mortality, and aetiologies of lower respiratory infections in 195 countries, 1990-2016: a systematic analysis for the Global Burden of Disease Study 2016. Lancet Infect Dis. 2018;18(11):1191-210.

5. Jadad AR, To MJ, Emara M, Jones J. Consideration of multiple chronic diseases in randomized controlled trials. JAMA. 2011;306(24):2670-2.

6. Adam L, Moutzouri E, Baumgartner C, Loewe AL, Feller M, M’Rabet-Bensalah K, et al. Rationale and design of OPtimising thERapy to prevent Avoidable hospital admissions in Multimorbid older people (OPERAM): a cluster randomised controlled trial. BMJ Open. 2019;9(6):e026769.

7. Man MS, Chaplin K, Mann C, Bower P, Brookes S, Fitzpatrick B, et al. Improving the management of multimorbidity in general practice: protocol of a cluster randomised controlled trial (The 3D Study). BMJ Open. 2016;6(4):e011261.

8. European Centre for Disease Prevention and Control. Seasonal influenza vaccination and antiviral use in EU/EEA Member States – Overview of vaccine recommendations for 2017–2018 and vaccination coverage rates for 2015–2016 and 2016–2017 influenza seasons. Stockholm: ECDC; 2018.

9. Bundesamt für Gesundheit, Eidgenössische Kommission für Impffragen. Schweizerischer Impfplan 2019. Richtlinien und Empfehlungen. Bern: Bundesamt für Gesundheit; 2019.

10. Top G, Carillo-Santisteve P. The vaccination programmes in Belgium. 2019. Available from: https://www.afmps.be/sites/default/files/content/08_implementation_vaccination_programmes.pdf.

11. Castiglia P. Recommendations for pneumococcal immunization outside routine childhood immunization programs in Western Europe. Adv Ther. 2014;31(10):1011-44.

12. Deshpande G, Visaria J, Singer J, Johnson KD. Impact of medical and/or pharmacy reimbursement on adult vaccination rates. Am J Manag Care. 2018;24(8 Spec No.):Sp286-sp93.

13. Santaularia J, Hou W, Perveen G, Welsh E, Faseru B. Prevalence of influenza vaccination and its association with health conditions and risk factors among Kansas adults in 2013: a cross-sectional study. BMC Public Health. 2016;16(1):185.

14. Kamal KM, Madhavan SS, Amonkar MM. Determinants of adult influenza and pneumonia immunization rates. Journal of the American Pharmacists Association. 2003;43(3):403-11.

15. Domínguez À, Soldevila N, Toledo D, Godoy P, Castilla J, Force L, et al. Factors associated with influenza vaccination of hospitalized elderly patients in Spain. PLOS ONE. 2016;11(1):e0147931.

16. Böhmer MM, Walter D, Müters S, Krause G, Wichmann O. Seasonal influenza vaccine uptake in Germany 2007/2008 and 2008/2009: results from a national health update survey. Vaccine. 2011;29(27):4492-8.

17. Dios-Guerra C, Carmona-Torres JM, Lopez-Soto PJ, Morales-Cane I, Rodriguez-Borrego MA. Prevalence and factors associated with influenza vaccination of persons over 65 years old in Spain (2009-2014). Vaccine. 2017;35(51):7095-100.

18. Zuercher K, Zwahlen M, Berlin C, Egger M, Fenner L. Trends in influenza vaccination uptake in Switzerland: Swiss Health Survey 2007 and 2012. Swiss Med Wkly. 2019;149:w14705.

19. Williams W, Lu P, O’Halloran A. Surveillance of Vaccination Coverage among Adult Populations — United States, 2015. MMWR Surveill Summ. 2017;66(No. SS-11):1–28.

20. Dubé E, Laberge C, Guay M, Bramadat P, Roy R, Bettinger JA. Vaccine hesitancy. Human Vaccines & Immunotherapeutics. 2013;9(8):1763-73.

21. Burns VE, Ring C, Carroll D. Factors influencing influenza vaccination uptake in an elderly, community-based sample. Vaccine. 2005;23(27):3604-8.

22. Mackenbach JP, Stirbu I, Roskam AJ, Schaap MM, Menvielle G, Leinsalu M, et al. Socioeconomic inequalities in health in 22 European countries. N Engl J Med. 2008;358(23):2468-81.

23. Schmedt N, Schiffner-Rohe J, Sprenger R, Walker J, von Eiff C, Hackl D. Pneumococcal vaccination rates in immunocompromised patients-A cohort study based on claims data from more than 200,000 patients in Germany. PLoS One. 2019;14(8):e0220848.

24. Hak E, Schönbeck Y, Melker HD, Essen GAV, Sanders EAM. Negative attitude of highly educated parents and health care workers towards future vaccinations in the Dutch childhood vaccination program. Vaccine. 2005;23(24):3103-7.

25. Castro-Sánchez E, Vila-Candel R, Soriano-Vidal FJ, Navarro-Illana E, Díez-Domingo J. Influence of health literacy on acceptance of influenza and pertussis vaccinations: a cross-sectional study among Spanish pregnant women. BMJ Open. 2018;8(7):e022132.

26. Stahl JP, Cohen R, Denis F, Gaudelus J, Martinot A, Lery T, et al. The impact of the web and social networks on vaccination. New challenges and opportunities offered to fight against vaccine hesitancy. Med Mal Infect. 2016;46(3):117-22.

27. Rehmet S, Ammon A, Pfaff G, Bocter N, Petersen LR. Cross-sectional study on influenza vaccination, Germany, 1999-2000. Emerg Infect Dis. 2002;8(12):1442-7.

28. Szucs TD, Muller D. Influenza vaccination coverage rates in five European countries-a population-based cross-sectional analysis of two consecutive influenza seasons. Vaccine. 2005;23(43):5055-63.

29. Sabapathy D, Strong D, Myers R, Li B, Quan H. Pneumococcal vaccination of the elderly during visits to acute care providers: Who are vaccinated? Preventive Medicine. 2014;62:155-60.

30. Greenland S. Interpretation and choice of effect measures in epidemiologic analyses. Am J Epidemiol. 1987;125(5):761-8.

31. Savitz DA. Measurements, estimates, and inferences in reporting epidemiologic study results. Am J Epidemiol. 1992;135(3):223-4.

32. Tamhane AR, Westfall AO, Burkholder GA, Cutter GR. Prevalence odds ratio versus prevalence ratio: choice comes with consequences. Stat Med. 2016;35(30):5730-5.

33. Barros AJD, Hirakata VN. Alternatives for logistic regression in cross-sectional studies: an empirical comparison of models that directly estimate the prevalence ratio. BMC Medical Research Methodology. 2003;3(1):21.

---

## [Decision Letter · Decision Letter 1]

3 Nov 2021

Cross-sectional study on the prevalence of influenza and pneumococcal vaccination and its association with health conditions and risk factors among hospitalized multimorbid older patients

PONE-D-21-21995R1

Dear Dr. Papazoglou,

We’re pleased to inform you that your manuscript has been judged scientifically suitable for publication and will be formally accepted for publication once it meets all outstanding technical requirements.

Kind regards,

Ray Borrow, Ph.D., FRCPath

Academic Editor

PLOS ONE

Reviewers' comments:

Reviewer's Responses to Questions

**Comments to the Author**

1. If the authors have adequately addressed your comments raised in a previous round of review and you feel that this manuscript is now acceptable for publication, you may indicate that here to bypass the “Comments to the Author” section, enter your conflict of interest statement in the “Confidential to Editor” section, and submit your "Accept" recommendation.

Reviewer #2: All comments have been addressed

2. Is the manuscript technically sound, and do the data support the conclusions?

Reviewer #2: Yes

3. Has the statistical analysis been performed appropriately and rigorously? 

Reviewer #2: Yes

4. Have the authors made all data underlying the findings in their manuscript fully available?

Reviewer #2: Yes

5. Is the manuscript presented in an intelligible fashion and written in standard English?

Reviewer #2: Yes

6. Review Comments to the Author

Reviewer #2: The revised version of the manuscript "Cross-sectional study on the prevalence of influenza and pneumococcal vaccination and its association with health conditions and risk factors among hospitalized multimorbid older patients" has been reviewed. The authors have adequately addressed my comments.

Some minor mistakes should be addressed in the section of References:

Reference 35: the capital letters of all the words in the title of the article (with exception of the first word) should be changed to small letters.

Reference 39: Because the name of the institution is written in French “belgium” should be changed to “Belgique”.

7. PLOS authors have the option to publish the peer review history of their article (what does this mean?). If published, this will include your full peer review and any attached files.

Reviewer #2: No

---

## [Editor Report · Acceptance letter]

5 Nov 2021

PONE-D-21-21995R1 

Cross-sectional study on the prevalence of influenza and pneumococcal vaccination and its association with health conditions and risk factors among hospitalized multimorbid older patients 

Dear Dr. Papazoglou:

I'm pleased to inform you that your manuscript has been deemed suitable for publication in PLOS ONE. Congratulations! Your manuscript is now with our production department. 

Kind regards, 

on behalf of

Prof. Ray Borrow 

Academic Editor

PLOS ONE